# Teaching Code Execution to Tiny Language Models

## Abstract

Recent advancements in large language models have demonstrated their effectiveness in various tasks. However, the question of these models' limitations remains open though. For instance, can a language model learn to perform code execution (i.e., predicting the output of code)? Current research indicates that the performance of state-of-the-art large language models in code execution is still limited. The reasons for this limitations are unclear though. Is it due to fundamental constraints or other factors such as training data and computational resources? Is the next-token prediction objective sufficient for learning code execution? How small can a language model be while still capable of learning code execution? In this paper, we investigate these questions. More specifically, we investigate whether tiny language models, trained from scratch using the next-token prediction objective, can effectively learn to execute code. Our experiments show that, given appropriate data, model size, and computational resources, tiny language models can indeed learn to perform code execution with a 99.13% accuracy for a tiny Turing-complete programming language. We begin by defining a tiny programming language called TinyPy. Millions of randomly generated codes in this language, along with their outputs, are used to train our tiny language models using the next-token prediction task. We then conduct a series of experiments to determine the smallest model size, data amount, and computational resources necessary to train our language model to achieve near-perfect accuracy in code execution. Our findings reveal that TEX, our proposed tiny language model with 15M parameters, can successfully learn code execution. This suggests that a task as complex as predicting code output is within the reach of language models.

## 1 Introduction

The rise of large language models (LLMs) such as GPT-3 (Brown et al., 2020) has marked a significant advancement in the field of natural language processing (NLP). These models, typically based on autoregressive, decoder-only architectures, have demonstrated superior performance across a broad range of language-related tasks. Building on this success, several studies have extended similar architectures to the domain of code modeling (Chen et al., 2021; Lu et al., 2021; Zheng et al., 2023), leading to the development of code language models trained on vast corpora of programming languages.

While these models have achieved state-of-the-art results in various coding tasks, such as code generation and completion, accurately executing code (predicting the output of a given code) remains a significant challenge for these models. Previous studies have highlighted this gap (Austin et al., 2021), indicating that even the largest code language models often struggle to perform code execution tasks reliably. In addition, our evaluation of state-of-the-art models, including Code Llama (Rozière et al., 2024) and GPT-4o, on our test dataset, reveals limitations in their ability to execute code perfectly (above 99% accuracy).

In this paper we investigate whether a language model can learn to execute code. Our investigation is driven by several key questions: Can a language model learn to predict the output of a code (i.e., learn code execution) accurately? Is the next-token prediction objective, which has been the cornerstone of language model training, sufficient for learning code execution? How can we effectively teach code execution to a language model? And perhaps most importantly, how small can a language model be while still maintaining the ability to learn and perform code execution tasks?

We believe that the task of code execution represents a significant challenge that offers valuable insights into the capabilities of language models. While code execution might not be inherently crucial for language models in practice, as dedicated tools exist for this purpose, it serves as an excellent proxy for assessing a model's ability to handle complex, structured information. The precision required for code execution make it an ideal benchmark for evaluating a language model's capacity to represent and manipulate abstract concepts, follow instructions and logical sequences, and produce accurate outputs based on specific commands and a formal syntax. By focusing on code execution, we aim to shed light on the capabilities and limitations of language models, potentially uncovering insights that could inform future developments in the field.

To answer our questions, we followed a structured experimental approach. We started by defining a tiny, yet turing-complete, programming language to train our tiny language model. We call it TinyPy. This language was specifically designed to be complex enough to maintain Turing-completeness while remaining minimalistic, allowing a tiny language model to learn and execute it. Next, we generated millions of random code snippets in TinyPy, creating a diverse dataset that covers various code structures and patterns. Following this, we trained models with different sizes from scratch using the next-token prediction objective. Finally, we evaluated these models on their ability to accurately execute code.

Our findings reveal that TEX, our proposed tiny language model with only 15M parameters, can achieve up to 99.13% accuracy in executing TinyPy code. Importantly, we find that the next-token prediction task is sufficient for teaching code execution to language models. By demonstrating that tiny models can learn to execute code, this study lays the groundwork for further exploration into the capabilities and applications of language models in code execution and beyond.

The main contributions of this work are as follows:

- We demonstrate that tiny language models, with less than 20M parameters, trained from scratch using only the next-token prediction objective, can achieve near-perfect accuracy (up to 99.13%) in code execution.
- We propose a tiny language model, TEX, that achieves 99.13% accuracy in the task of TinyPy code execution.
- We conduct a systematic analysis of the key factors —data size, model size, and training duration—that directly influence the performance of a language model in executing code.
- We release our dataset, and open-source our random code generator, model, and entire codebase, enabling the replication and extension of our work by the research community.

## 2 RELATED WORK

**Code execution.** Previous research has extensively explored the task of code execution, highlighting its importance. Various architectures have been employed for this purpose, including recurrent neural networks (RNNs) (Zaremba & Sutskever, 2015), graph neural networks (GNNs) (Bieber et al., 2020; Wang et al., 2020), and Transformers (Dehghani et al., 2018; Yan et al., 2020). They all take a piece of code as input and predict its output.

Recent studies have also leveraged pre-trained models to perform the task of code execution. For instance, Austin et al. (2021) evaluated pre-trained models, ranging in size from 2M to 137B parameters, and found that even the largest models struggle to predict program outputs, even with fine-tuning, never exeeding a 29% accuracy. Nye et al. (2021) introduced a "scratchpad" to store intermediate computation steps, enhancing the ability of models to perform multi-step computations. A study done by Liu et al. (2023b) investigates how well pre-trained models can perform code execution. Their model, CodeExecutor, is an encoder-decoder model that uses a training objective designed specifically for the task of code execution and which involves predicting both the line order and the intermediate states of the execution trace. With this specifically designed pre-trained task, CodeExecutor could reach an accuracy of $48.06\%$ in predicting outputs. While this result represents a significant milestone, current state-of-the-art work still does not answer the question of whether language models can perfectly learn the task of code execution? Is this task within the reach of current language models, given enough training data? Or even with a large amount of data and training time, language models cannot perform this task? Is a generic pre-training objective such as next token-prediction enough, or we need to use a specialized training objective? Even state-of-the-art

large language models such as Code Llama (Rozière et al., 2024) and GPT-4o still have limitations in this task as we show in the evaluation. Does this mean that the task itself is fundamentally beyond the capabilities of language models, or simply it is a matter of lack of data and lack of training time? All of these questions are still open research questions. Our focus in this paper is to address these questions rather than simply building a model for code execution. We belive that the task of code execution is a challenging task that provides insights on the abilities of language models and therefore we use it to study the capabilities of language models.

To the best of our knowledge, our work is the first to demonstrate that a language model pre-trained using next-token prediction, can indeed learn code execution with near-perfect accuracy (99.13%). That is, given enough data, and training time, language modeling alone is enough to learn the task of code execution, assuming the output of code is present in the training data. Unlike previous wok, we focus on studying decoder-only language models, and we do not define a specific objective for this task but rather use the generic next-token prediction objective, known as language modeling, which is a widely used objective for language models.

**Small language models.** Recent advancements in small language models have revealed their potential across various specialized tasks. Lee et al. (2023) demonstrates that small transformers can efficiently learn to perform arithmetic operations when trained on high-quality data. Eldan & Li (2023) illustrate the capability of small models with fewer than 10 million parameters to generate fluent and coherent short stories in natural language. Liu et al. (2023a) highlight that fine-tuning small models on high-quality datasets can achieve high accuracy on the GSM8K benchmark. Additionally, Joshi et al. (2024) present a 60M parameter model trained on Excel formulas that outperforms larger models in tasks related to Excel formulas. Our work extends this line of research by being the first to thoroughly explore the performance of small language models on the task of code execution and whether it can be learned by language models.

## 3 OVERVIEW

Our study employs an incremental approach to investigate code execution capabilities in tiny language models. We start by designing a tiny Turing-complete programming language, called TinyPy. This language is tiny because it supports minimal language constructs that allow it to be turing-complete but does not support more complex language constructs (which are usually designed to improve productivity). The language has a vocabulary size of 53 tokens, supports statements that perform arithmetic operations, and control flow (if-conditionals, for loops and while loops). It supports 8-bit unsigned integers as a data type (more detail about TinyPy in subsection 4.1). Next, we use the TinyPy Generator (Yamani et al., 2024), an automatic python code generation tool, to generate millions of codes written in this language (more on data generation in subsection 4.2). The dataset consists of the randomly generated codes, each followed by its corresponding output expressed in a comment at the end of the code. We use this dataset to train our tiny language models.

Through iterative refinement, we develop our proposed model, **TEX (Tiny Executor)**, a language model of 15M parameters that achieves 99.13% accuracy in code execution (details about the model are provided in section 5). This result demonstrates that a language model (as tiny as TEX) can effectively learn to execute code when trained from scratch using the next-token prediction only. Figure 1 provides an overview of our approach.

## 4 DATA

The majority of code language models are trained on code sourced from publicly repositories available on the internet. While such an approach has advantages, it also creates challenges without being necessary in our context. First, because our main objective is to answer the fundamental question of whether a language model can learn the task of code execution for a Turning-complete language, the language itself does not need to be complex. The simplest Turing-complete language is sufficient. Second, given our objective of training a tiny language model, utilizing a complex programming language would be inefficient due to its complexity and large vocabulary. To address these challenges, we designed our own tiny yet Turing-complete language, which is detailed in subsection 4.1.

Furthermore, it is critical that the training data consists of executable codes. This requirement is difficult to guarantee when using code collected from public repositories, as such code often requires

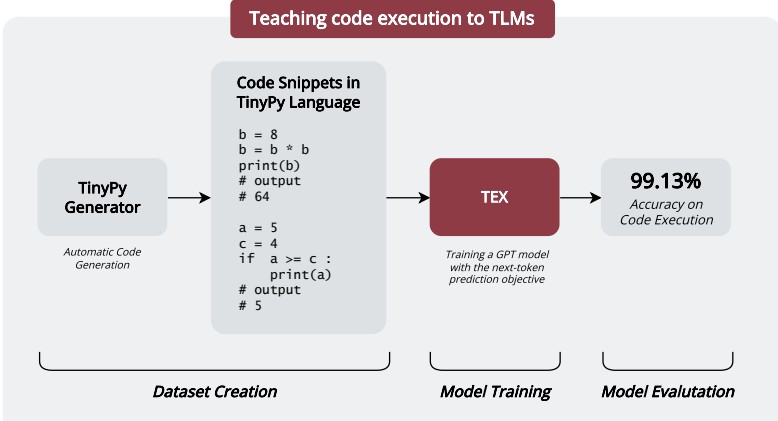

Figure 1: Overview of our Approach

complex dependencies and is not easily executable. For example, to execute a function extracted from a public repository of a large project, one needs to initialize the context and all the data structures necessary to run that function. Meaningful data needs to be passed as input to the function, in addition to a correct initialization of all the global variables, which are all hard to automate on large scale. This is a well-known research problem in the programming language and compiler community with on-going efforts to build datasets of executable code for different languages (da Silva et al., 2021). Therefore, to ensure that our code is executable, we generated our own dataset of executable code, which we describe in detail in subsection 4.2.

### 4.1 LANGUAGE DESCRIPTION

The custom language we developed, named TinyPy, is a Turing-complete language derived from a carefully selected subset of Python. This subset was chosen to retain the essential language constructs of Python that allow for Turing-completeness while minimizing complexity, making it suitable for our tiny language model.

TinyPy has the following language constructs (full grammar is provided in Appendix A) :

- Assignment statements;
- Arithmetic operations (addition, subtraction, multiplication, and division);
- Comparison operators ($\leq, <, \geq, >, =, \neq$) and the logical operator $not$
- Control flow:
    - Conditional statements (if-else, if-elif-else);
    - Loops (for loops and while loops).

TinyPy is a Turing-complete language due to its support for 1) sequences of statements; 2) conditionals (if-then-else); and 3) while loops. According to the Böhm-Jacopini theorem (Böhm & Jacopini, 1966), also known as the structured program theorem, these three language constructs are sufficient to make a given language Turing-complete.

To further limit the problem space, we've restricted the numeric range of values used for initializing variables to unsigned 8-bit integers. Note that we do not put restrictions on the range or data type of the intermediate variables. Such variables can be floating point numbers and can be negative. The final output can also be a floating point number or a negative number.

Note also that, the fact that TinyPy supports only 4 arithmetic operations and unsigned 8-bit integers (for data initialization) does not make it Turing-incomplete. It is still Turing-complete, but has a lower level of abstraction. Moreover, complex operators and data types can still be emulated using the basic operators and data type supported by TinyPy (for example, one can encode a floating point number using a sequence of the two numbers 0 and 1).

When designing our language, we had three goals in mind. First, the language should allow us to train language models on a small vocabulary compared to complete programming languages. This would allow us (as well as future researchers in the community) to perform experiments on

limited resources. Second, the language should have enough expressive power to allow for important programming patterns. Third, the language should have the same syntax as an existing language, so that we avoid the need to build a while new compiler and software ecosystem. TinyPy satisfied these goals.

## 4.2 DATASET GENERATION

To ensure that the generated code is both syntactically correct and executable within the constraints of our custom language, we utilized TinyPy Generator, an automated Python code generation tool proposed by Yamani et al. (2024). This tool leverages a context-free grammar to produce synthetic Python programs that strictly adhere to the predefined syntax of TinyPy. In addition to generating code snippets, TinyPy Generator also executes each program and records both the code and its output (expressed in a comment at the end of the code) in a structured file format. Figure 2 shows examples of codes generated by TinyPy Generator and that we use in our dataset.

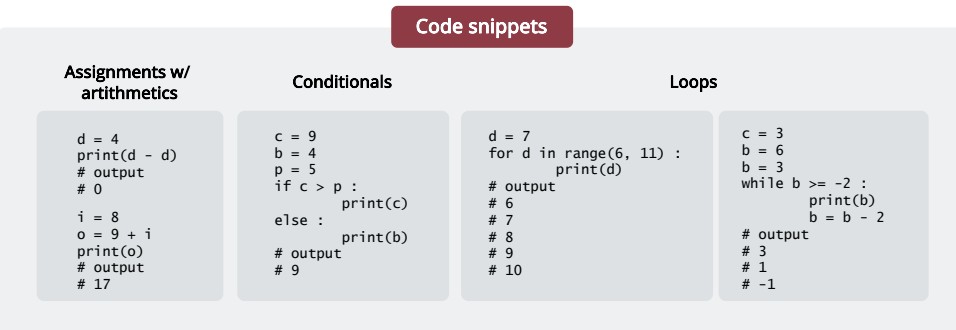

Figure 2: Code Snippets in TinyPy generated by TinyPy Generator (Yamani et al., 2024).

## 4.3 DATA DEDUPLICATION

To prevent the generation of duplicate programs, the TinyPy Generator (Yamani et al., 2024) utilizes a hash function to create a unique hash for each generated code. This hash is then compared against those of existing codes in the dataset. If a match is detected, indicating that the new program is identical to an existing one, the duplicate is discarded, ensuring that only unique codes are retained.

## 5 MODEL

We use on the GPT (Generative Pre-trained Transformer) architecture for this study, originally introduced by Radford et al. (2019). Specifically, we use NanoGPT, a small-scale, open-source implementation inspired by GPT-2, developed by Karpathy (2022). NanoGPT provides flexibility for training from scratch (random initialization) and customization to meet specific research requirements. For our experiments, we configure the model with six transformer layers and attention heads (Figure 3), and omit biases in the linear layers

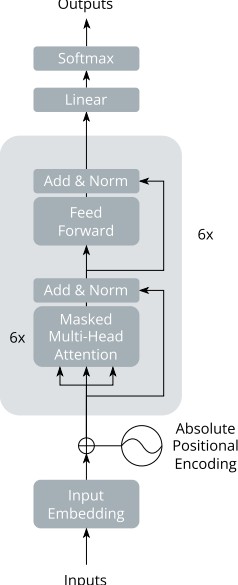

Figure 3: NanoGPT Architecture, inspired from (Vaswani et al., 2017).

# 6 EXPERIMENTS

## 6.1 EXPERIMENTAL SETUP

**Training.** All models are trained from a random initialization (i.e., from scratch) using the conventional next-token prediction objective. No dropout is applied, with a batch size of 64 and a block size of 256. We use character-level tokenization and absolute position encoding. The learning rate is set to 1e-3, and the AdamW optimizer is employed. Learning rate decay is implemented at milestones of 70%, 80%, and 90% of the total iterations. All our models were trained on a single Quadro RTX 8000 GPU.

**Evaluation Methodology.** To investigate the code execution capabilities of our models, we evaluate them on our test set. Each model is prompted with the code portion of the code snippets, stopping at the '# output' comment to exclude the actual output and allow the model to predict it. The model generates the output token by token, as follows: First, the model is provided with the context (as shown in Figure 4). The model then predicts the logits for the next token based on this context. These logits are converted into a probability distribution via softmax. The `torch.multinomial` function is used to sample the next token from this distribution with a temperature of 0.4. This sampled token is added back to the context. This procedure is repeated until the maximum number of new tokens has been generated. The final output consists of all the tokens generated by the model.

**Evaluation Metrics** We use the Output Accuracy as the evaluation metric, which checks if the generated output exactly matches the expected output from running the code.

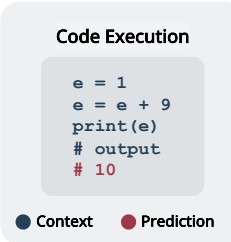

Figure 4: Code Execution Evaluation Task

## 6.2 EVALUATING OUR PROPOSED MODEL (TEX)

Based on extensive experimentation with various configurations of model size, data size and training duration (detailed in sections 6.4, 6.5, and 6.6), we developed TEX (Tiny Executor), a 15M-parameter decoder-only model. TEX was trained on a dataset containing 96 million code snippets over the course of 5 epochs which is equivalent to 5 days and 18 hours of training. TEX achieves an **accuracy of 99.13%** on the task of code execution. This result is particularly noteworthy given the model's compact size and generic training objective, showing that even tiny language models, can indeed learn to perform the task of code execution accurately.

## 6.3 EVALUATING LLMS ON CODE EXECUTION

To illustrate the complexity of the code execution task and the challenges faced by current advanced models in predicting the output of a given code snippet, we evaluated several models using our test set (the same one used for evaluating TEX). We asked each model to execute the code and provide the result as a comment, using a single example as a prompt, which is considered 1-shot learning. For instance, the prompt would look like this:

> Run the code snippet and append the results as a comment. For example:
> Prompt:
>
> ```
> a = 1
> b = 1
> print(a + b)
> # output
> ```
>
> The answer should be: # 2.

The results of this evaluation are shown in Table 1.

| Model | Access | Code Execution Accuracy |
|---|---|---|
| Code LLaMa 13B (Rozière et al., 2024) | Open Source | 16.18% |
| LLaMa 3.2 3B | Open Source | 39.18% |
| GPT-4o-mini | Closed Source | 75.12% |
| GPT-4o | Closed Source | 87.37% |
| TEX (ours) | Open Source | **99.13%** |

Table 1: Comparison of LLM performance on code execution

These results highlight the limitations of state-of-the-art LLMs in performing code execution tasks, even in a simplified setting. Despite testing them on code snippets written in TinyPy, which can be seen as a simple subset of the python programming language, state-of-the-art models still have limitations in predicting code output.

## 6.4 DATA SIZE SCALING

This experiment evaluates the relationship between the size of the training data used to train our model and its performance, measured by accuracy in predicting the code output. Models with 1M parameters were trained by fixing the embedding dimension to 120 for 1.7 epochs. The size of the training dataset was scaled from 2 million to 64 million. Results are presented in Figure 5.

- With the smallest data size of 2 million snippets, the model achieves an accuracy of 63.18%, indicating that limited data leads to suboptimal performance.

- Increasing the data size to 4 million snippets results in a significant improvement, with the accuracy rising to 77.85%. This demonstrates the model's ability to leverage additional data for better generalization.

- Further increases in the data size continue to enhance accuracy, with 8 million snippets yielding 84.02% accuracy and 16 million snippets reaching 86.95%. These results suggest diminishing but consistent returns from adding more data.

- Beyond 16 million snippets, the accuracy gains start to taper off. For 32 million snippets, the accuracy climbs modestly to 87.91%, followed by 90.41% for 48 million snippets.

- Finally, the largest data size of 64 million snippets results in a slight improvement to 90.64%, indicating that the model is nearing its performance ceiling with respect to the data size used.

Figure 5 clearly shows that increasing data size improves model accuracy, though the rate of improvement slows down as the data size grows larger. This points to a potential saturation point where additional data yields diminishing returns, which is important for optimizing resource allocation when scaling data for model training. Note that these results are valid for a model size of 1M parameters. Larger models will likely be able to benefit from more data. Studying what is the best amount of data for *any* given model size is not within the scope of this experiment (and our work in general). Our focus in this experiment is rather to explore the effect of increasing the training data on performance for a given tiny model (we chose a size of 1M in the experiment).

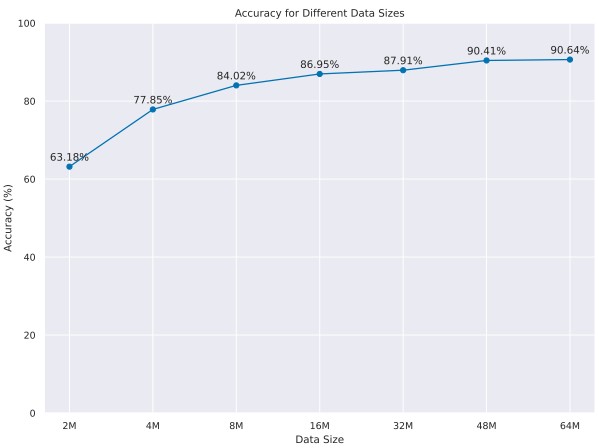

Figure 5: Data Size Scaling Accuracy

## 6.5 MODEL SIZE SCALING

This experiment evaluates the relationship between the model size and accuracy in the task of code execution. Models were trained for 1.7 epochs on a dataset comprising 48 million code snippets. The model size was systematically scaled from 100K to 20M parameters by adjusting the embedding dimension.

The results demonstrate a significant correlation between model size and accuracy in code execution tasks. As depicted in Figure 6, a clear trend of increasing accuracy with larger model sizes is observed.

- Starting with the smallest model size at 100K parameters, the accuracy is notably low at 34.32%. This suggests that minimal parameterization is insufficient for capturing the complexity required for accurate code execution.

- As the model size increases to 0.5M parameters, we notice a significant increase in accuracy to 85.80%, highlighting the importance of a larger number of parameters for learning more intricate patterns in code.

- The accuracy incrementally improves as the model size scales up, with 2M parameters reaching an accuracy of 90.06%, and 5M parameters slightly enhancing to 93.08%. This progression underscores a diminishing return on accuracy gains as the model size expands.

- The trend continues subtly with 10M and 15M parameter models, achieving accuracies of 93.86% and 94.38% respectively. The relatively small improvements suggest approaching an asymptote, where additional parameters contribute less significantly to model performance.

- The largest model size tested, 20M parameters, reaches the peak accuracy of 94.58%, affirming the correlation between model size and performance up to a certain threshold beyond which gains may plateau.

Figure 6 clearly indicates that while larger models generally perform better in terms of accuracy, the efficiency of scaling in terms of parameter increase versus performance gain should be considered, particularly when computational resources or training time are limiting factors.

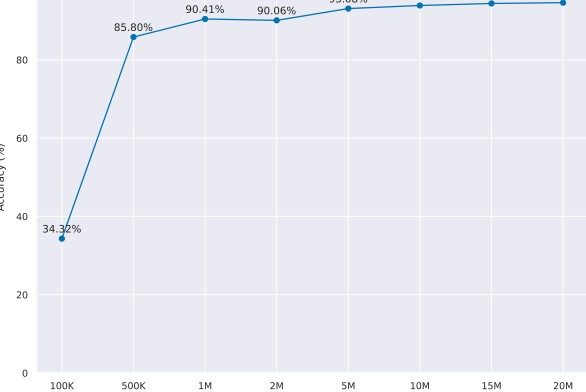

Figure 6: Model Scaling Accuracy

## 6.6 COMPUTE AMOUNT SCALING

This experiment evaluates the relationship between the amount of compute (number of epochs) and accuracy in the task of code execution. Models with 1M parameters were trained on 48 million code snippets. The compute amount was scaled by varying the number of epochs from 0.7 to 8.

The results in Figure 7 demonstrate the effect of scaling the number of training epochs on model accuracy for a fixed model size of 1M parameters and a fixed data size of 48 million code snippets.

- The experiment starts with 0.7 epochs, resulting in an accuracy of 85.36%. This lower accuracy suggests insufficient training time, which limits the model's ability to fully learn from the data.

- As the number of epochs increases to 1 and 1.7, there is a rapid improvement in accuracy to 88.86% and 90.41%, respectively. This shows the benefits of additional training time early in the process.

- The accuracy gains continue with more epochs, reaching 90.53% at 3 epochs and 91.45% at 4 epochs, but at a slower rate, indicating diminishing returns from increased training.

- The highest accuracy of 92.39% is achieved at 5 epochs, marking the optimal point for this configuration. Beyond this, further increases in epochs do not lead to significant improvements.

- Interestingly, at 6 epochs, the accuracy dips slightly to 90.81%, possibly due to overfitting or other model dynamics.

- Accuracy improves again to 91.92% and 92.25% at 7 and 8 epochs, suggesting some recovery, though the gains are marginal compared to the peak at 5 epochs.

Figure 7 highlights the importance of selecting an appropriate number of epochs for training. While more epochs generally improve model performance, there is a trade-off, as excessive training can lead to diminishing returns or even overfitting.

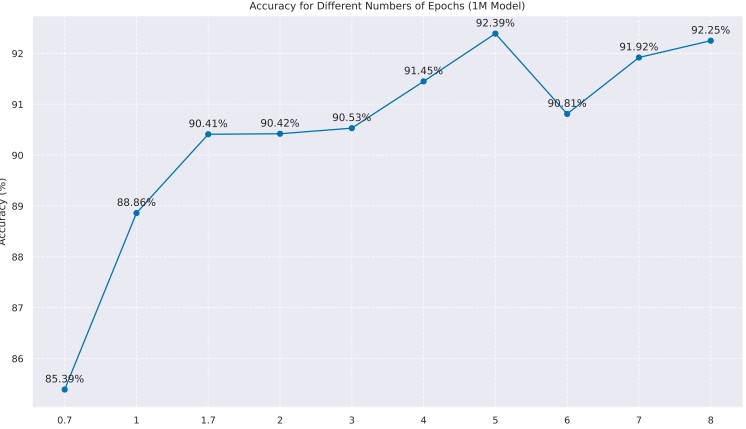

Figure 7: Compute Scaling Accuracy

# 7 DISCUSSION

Our experiments demonstrate that tiny language models, such as the 15M-parameter TEX model, can achieve high accuracy (up to 99.13%) in the task of code execution on a simple, yet Turning-complete language such as TinyPy. The results also highlight that the task of next-token prediction is sufficient to teach a model the task of code execution, assuming the training data contains enough samples of codes and their outputs.

**Next-Token Prediction for Code Execution.**   Our research provides a proof-of-concept that the next-token prediction objective, commonly used in language model training, is indeed sufficient for learning the task of code execution. The high accuracy achieved by our model, trained solely using next-token prediction on data containing codes and their output, without any other special type of training, demonstrates the effectiveness of next-token prediction in capturing the underlying patterns and logic of code execution. This finding is significant as it suggests that, unlike the current common practice in state-of-the-art models for code execution (Liu et al., 2023b), specialized training objectives may not be necessary for teaching code execution to language models.

**Scaling Experiments.**   Our study reveals that teaching code execution to language models requires a careful balance between data size, model size, and training duration. We observed diminishing returns beyond certain thresholds— 15M parameters for model size, and 5 epochs for training. However, these results are specific to our experimental configuration, and different data sizes, model capacities, or training durations may yield different outcomes. Studying what is the best amount of data, model size and training time, is not within the scope of this work. Further exploration is needed to perform a comprehensive exploration across a broader range of configurations.

**Minimum Model Size for Code Execution.**   Our research provides insights into the question of how small a language model can be while still maintaining the ability to learn and perform the task of

code execution. The near-perfect performance of our 15M-parameter model suggests that even tiny models can effectively learn code execution. However, our scaling experiments indicate that there is a lower bound to model size, below which performance degrades significantly. For our specific task and language, models with fewer than 1M parameters struggled to achieve high accuracy.

**Language Limitations**  It is important to note that the limitations of the TinyPy langauge are orthogonal to our main hypothesis. While our designed language may have a lower level of abstraction and may lack certain features found in programming languages with a higher level of abstraction, these limitations do not impact our core investigation, as the language is complex enough and has the minimum programming language constructs to qualify as Turing-complete (sequences of statements, for and while loops and if-conditionals). Our primary focus in this work is on the fundamental ability of language models to learn and execute a simple form of code (yet complex enough). Now that we have demonstrated such an ability, we plan to extend our work to support full programming languages (e.g., Python).

## 8  LIMITATIONS

While our study demonstrates the potential of tiny language models in code execution tasks, it has some limitations. Our use of the TinyPy, our custom programming language, allowed us to focus on core programming language constructs but does not cover more complex constructs. Further studies are needed to generalize our findings to other, more complex programming languages. Such studies are orthogonal to our goal in this paper though. Our goal is to evaluate whether a language model can learn the task of code execution on a simple, yet turing-complete language, and TinyPy is sufficient to achieve this goal. Additionally, the restricted numeric range of unsigned 8-bit integers limits the scope of numerical computations addressed, potentially overlooking challenges present in broader numerical ranges. These limitations provide valuable directions for future research, inviting exploration of the capabilities of tiny language models in more diverse and complex programming environments, and investigation of their performance across a wider range of programming languages and numerical operations.

## 9  CONCLUSION

This study provides empirical evidence that tiny language models, with fewer than 20M parameters, can effectively learn to execute simple code with high accuracy. Our proposed model TEX achieves a 99.13% accuracy in executing code from TinyPy, a Turing-complete programming language that performs arithmetic operations, and supports control flow statements such as if-conditionals, for loops and while loops.

The development of TEX, a 15M parameter model, serves as a proof-of-concept that a language model trained from scratch using only the next-token prediction objective on data containing codes and their outputs, and without any other task-specific training objective, can effectively learn to execute code.

This research contributes to the understanding of language models' capabilities in executing code and opens new directions for efficient and accessible AI models in coding applications. All code and datasets used in this study are open-sourced, providing a foundation for further research and development in this area.

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

648
649

# A TINYPY'S COMPLETE SET OF PRODUCTION RULES

650
651

## A.1 BASIC LANGUAGE CONSTRUCTS

652
653

This set of rules defines the basic language constructs such as variables, digits, arithmetic operators, relational operators, logical operators, and some special characters and keywords.

654
655
656
657
658
659
660
661
662
663
664
665
666
667
668
669
670
671
672
673
674

```
<variable> ::= a | b | c | ... | z
<digit> ::= 0 | 1 | 2 | ... | 9

<arithmetic_operator> ::= + | - | * | /
<relational_operator> ::= < | > | <= | >= | != | ==
<logical_operator_infix> ::= and | or
<logical_operator_prefix> ::= not

<new_line> ::= \n
<tab_indent> ::= \t
<bracket_open> ::= (
<bracket_close> ::= )
<equals> ::= =
<colon> ::= :
<comma> ::= ,
<if> ::= if
<elif> ::= elif
<else> ::= else
<for> ::= for
<in> ::= in
<range> ::= range
<while> ::= while
<print> ::= print
```

675
676

## A.2 EXPRESSIONS AND ASSIGNMENTS

677
678
679

This set of rules defines how expressions, enclosed expressions, display expressions, and assignments are formed.

680
681
682
683
684
685
686
687
688
689
690
691
692
693
694
695
696
697
698

```
<term> ::= <expression_identifier> | <digit>
<expression> ::= <term> <space> <operator> <space> <term>
<enclosed_expression> ::= <bracket_open> <expression>
    <bracket_close>
<display_expression> ::= <expression_identifier> <space>
    <operator> <space>

<expression_identifier> | <expression_identifier> <space>
    <operator> <space> <digit>
<identifier_initialization> ::= <identifier_initialization>
    <initialization> | <initialization>
<initialization> ::= <variable> <space> <equals> <space> <digit>
    <new_line>

<simple_assignments> ::= <variable> <space> <equals> <space>
    <expression> <new_line> | ""
 <new_line> | <variable> <space> <equals> <space> <expression>
     <new_line> | ""
<simple_arithmetic_evaluation> ::=
    <simple_arithmetic_evaluation> <arithmetic_operator>
    <enclosed_expression> | <enclosed_expression>
```

699
700

## A.3 CONDITIONAL STATEMENTS

701

This set of rules defines the formation of simple and advanced conditional statements (if, elif, else).

```
        <simple_if_statement> ::= <if> <space> <condition> <space>
            <colon> <new_line>
        <simple_elif_statement> ::= <elif> <space> <condition> <space>
            <colon> <new_line>
        <else_statement> ::= <else> <space> <colon> <new_line>
        <condition> ::= <optional_not> <condition_expression> |
            <condition_expression>
        <condition_expression> ::= <expression_identifier> <space>
            <relational_operator> <space> <expression_identifier> |
            <expression_identifier> <space> <relational_operator>
            <space> <digit>
        <optional_not> ::= <logical_operator_prefix> <space> | <space>
```

## A.4 Loop Constructs

This set of rules defines the formation of for and while loops.

```
        <for_header> ::= <for> <space> <expression_identifier> <space>
            <in> <space> <range> <bracket_open> <initial> <comma>
            <space> <final> <comma> <space> <step> <bracket_close>
            <space> <colon> | <for> <space> <expression_identifier>
            <space> <in> <space> <range> <bracket_open> <initial>
            <comma> <space> <final> <bracket_close> <space> <colon>
        <initial> ::= <digit>
        <final> ::= <step> * <execution_count> + <initial> – 1
        <step> ::= 1 | 2 | 3
        <execution_count> ::= 2 | 3
        <for_loop> ::= <for_header> <new_line> <tab_indent> <display>

        <while_header_less> ::= <while> <space>
            <condition_expression_less> <space> <colon> <new_line>
        <while_loop_less> ::= <while_header_less> <tab_indent> <display>
            <new_line> <tab_indent> <update_less>
        <update_less> ::= <while_identifier> <space> <equals> <space>
            <while_identifier> <space> <plus> <space> <step>

        <while_header_greater> ::= <while> <space>
            <condition_expression_greater> <space> <colon> <new_line>
        <while_loop_greater> ::= <while_header_greater> <tab_indent>
            <display> <new_line> <tab_indent> <update_greater>
        <update_greater> ::= <while_identifier> <space> <equals> <space>
            <while_identifier> <space> <minus> <space> <step>
```

## A.5 Display and Levels

This set of rules defines how print statements are formed and how different levels of language constructs are combined.

```
        <display> ::= <print> <bracket_open> <display_identifier>
            <bracket_close>
        <advanced_display> ::= <display> | <print> <bracket_open>
            <display_expression> <bracket_close>

        <level1> ::= <identifier_initialization> <simple_assignments>
            <advanced_display>
        <level2> ::= <identifier_initialization> <simple_if_statement>
            <tab_indent> <display> | <identifier_initialization>
            <simple_if_statement> <tab_indent> <display> <new_line>
            <simple_elif_statement> <tab_indent> <display> <new_line>
            <else_statement> <tab_indent> <display> |
            <identifier_initialization> <simple_if_statement>
```

```
        <tab_indent> <display> <new_line> <else_statement>
        <tab_indent> <display>
<level3> ::= <identifier_initialization> <for_loop>
<level4> ::= <identifier_initialization> <while_loop_less> |
    <identifier_initialization> <while_loop_greater>

<all> ::= <level1> | <level2> | <level3> | <level4>
```

