# OpenReview forum: "Teaching Code Execution to Tiny Language Models"
_ICLR.cc/2025/Conference — Submitted to ICLR 2025_

### Official Review · Reviewer_XpA7 · 2024-11-03

**Soundness:** 3
**Presentation:** 2
**Contribution:** 1
**Rating:** 1
**Confidence:** 4

**Summary:**

The authors define a subset of the Python programming language TinyPy, and randomly sample sequences from this subset of Python. The authors fine-tune a NanoGPT model of 15M parameters to predict the output of programs sampled from this simple language.  The authors are able to achieve a 99.13% accuracy on this task of predicting outputs. The authors then evaluate how much training data / model parameters are required to achieve this level of performance.

**Strengths:**

The authors are clear in their writing of the paper: ideas are expressed clearly. The authors do indeed highlight a specific research question: can models achieve a high accuracy on predicting TinyPy outputs, and how do parameter sizes / training corpus sizes affect performance.

**Weaknesses:**

The research questions posed by the paper do not meet the threshold of novelty and significance for a conference like ICLR. There are also some issues in terms of the presentation of the paper as well.

Merit/Novelty of Approach: It has been commonly done in the past in the ML literature and also in the Compiler literature to sample programs by sampling from the grammar of a programming language. Much early ML program synthesis literature used Karel, and would sample executable programs from this language e.g. https://arxiv.org/pdf/1805.04276. To my understanding as well, I did not see any part of the paper that guaranteed that the TinyPy programs would print out outputs to the console: I am not assured that the programs are non-trivial to predict outputs for.

Merit of Findings: It is generally accepted that more data and larger models will yield better performance in tasks such as program synthesis. I do not find it insightful that such small models can achieve such high accuracy on the program synthesis task in question without being provided more insights into more complex questions / problems in program synthesis or LLM reasoning.

Concerns about anonymity: The authors use term TinyPy to describe their language for program synthesis. This happens to share the same name as the TinyPy Generator (Yamani et al., 2024) they cite and claim to use. I am concerned the authors did not take sufficient efforts to de-anonymize their submission. E.g. a different name instead of TinyPy could have been adopted.

I think the exposition of the work and questions asked seems generally sound, and is great! But I would encourage the authors to review more literature in the areas of program synthesis and in reasoning and think about unanswered research questions that will be of greater interest. The work is executed well! But I think the directions/questions asked could be improved, and being more familiar with current important questions and problems in the field is something that may be important to focus on next!

**Questions:**

1. Do you have statistics for how the minimum and average number of print statements there are per program in the test set? Likewise for loop-constructs, what is the average number of loop constructs for the test set? Providing statistics on these could help better-clarify the difficulty of the task.
2. Have I made any misrepresentations in the comments I have made?

---

> ### Author Response · Authors · 2024-11-24
>
> Thank you for your thorough review and insightful comments. We appreciate the time and effort you've put into evaluating our work. Before addressing your specific points, we'd like to clarify an important aspect of our methodology:
>
> Correction on Training Approach:
> In your summary, you mentioned that we "fine-tune a NanoGPT model of 15M parameters." We want to emphasize that this is not accurate. Our model is trained from scratch, not fine-tuned. This is an important distinction as it demonstrates that our tiny language model can learn code execution capabilities without relying on pre-existing knowledge from a larger pre-trained model.
>
> Now, we'd like to address your questions and concerns point by point:
>
> 1. Statistics on print statements and loop constructs:
> We appreciate your request for more detailed statistics about our test set. As detailed in the appendix, our grammar defines four levels of code complexity (<level1> | <level2> | <level3> | <level4>), with an equal probability of generating each level. Levels 3 and 4 specifically correspond to 'for' and 'while' loops respectively. Given this structure, approximately 50% of the programs in our test set contain loop constructs. Additionally, every code snippet in our dataset is guaranteed to have one print statement.
>
> 2. Addressing your concerns:
>
> a) Merit/Novelty of Approach:
> We thank you for bringing attention to the paper on program synthesis. While there are similarities in using generated programs, our work is distinct in its focus and methodology. The paper you mentioned focuses on program synthesis using reinforcement learning and sequence-to-sequence models, addressing issues like program aliasing and syntactic correctness. In contrast, our work investigates code execution capabilities of tiny language models using next-token prediction, with a specific focus on scaling studies and demonstrating that small models can learn to execute code accurately.
> Regarding the concern about output generation, we apologize for not making this clear in the paper. All TinyPy programs in our dataset are guaranteed to produce console output, as our code generation process ensures at least one print statement in each program.
> The non-triviality of our TinyPy programs is clearly demonstrated by the results in Table 1 of our paper. This empirically shows that predicting the outputs of our TinyPy programs is indeed a challenging task, even for much larger models than the ones we primarily focus on in our study.
>
> b) Merit of Findings:
> We appreciate your perspective on the relationship between data, model size, and performance. However, we'd like to emphasize that our study's primary contribution isn't merely demonstrating improved performance with more data and larger models, but rather showing that tiny language models can achieve near-perfect accuracy in code execution using only next-token prediction.
> Regarding the suggestion to consider "more complex questions/problems", we believe this is premature given that current large language models struggle with our existing test set, as evidenced by their poor performance shown in Table 1.
>
> c) Concerns about anonymity:
> We appreciate your concern about anonymity. The use of "TinyPy" in our paper refers to the language defined by the TinyPy Generator tool, which is a publicly available resource. We chose to use this name for clarity and consistency with the tool we utilized, rather than creating a new name that might cause confusion. Our intention was not to compromise anonymity but to provide accurate references to the tools and resources used in our research.
>
> We hope these clarifications address your concerns and highlight the novelty and significance of our work. We're grateful for your suggestions to review more literature in program synthesis and reasoning, and we'll certainly take this advice into account for our future research directions.
>
> Thank you again for your valuable feedback. We believe our work contributes meaningful insights to the field, particularly in understanding the capabilities of tiny language models in complex tasks like code execution.

---

> > ### Comment · Reviewer_XpA7 · 2024-12-02
> >
> > Thanks for trying to clarify the perceived contributions you have.
> >
> > ## On Merit and Novelty
> >
> > I can understand and I can agree that the research question here is whether or not tiny-LMs can learn to execute programs. And I think this is a good framing. I strongly believe that the contribution does not meet the threshold for a conference like ICLR, but I think it is good you tried to submit the paper. I also appreciate the soundness of your experiments. The effort and implementation is good, and I would recommend submitting this paper to a workshop.
> >
> > One thing I can say, albeit cliché, is that "neural networks are universal function approximators." We sort of "in theory" know that with enough examples (e.g. randomly sampling programs), that with enough neural model capacity these sort of things should be able of being learned. Put another way, I do not walk away from this paper feeling like I really learned anything I can take away, because the nature of the simplified programming language may be so far removed from complex programming tasks, that it is hard for me to extrapolate anything meaningful. Indeed there are papers that explore phenomena with toy-like datasets, but often those papers may have either been from a "different era" or the research questions are framed to be of higher intellectual consequence. I agree that the fact it can handle looping constructs and that these make up ~50% of the dataset is interesting, but more analysis of this and/or how it is of consequence can benefit from more work.
> >
> > I would really just place the rating I have in terms of the choice of research direction and topic, not the implementation. I look forward to the authors' future work, as they are clearly capable of sound and thorough research. Your experimental implementation and clear documentation of the results shows strong research fundamentals. While this particular work may be better suited for a workshop, the ability to carry out sound research is clear here.

---

### Official Review · Reviewer_zxGo · 2024-11-03

**Soundness:** 3
**Presentation:** 3
**Contribution:** 3
**Rating:** 3
**Confidence:** 4

**Summary:**

This paper investigates whether LLMs can learn to predict the code execution results by designing a simplified turing-complete python-like programming language, generate codes in this language, and train tiny LMs from scratch using next-token prediction.

**Strengths:**

The problem investigated in this paper is interesting. Nowadays, LLMs are indeed strugging to predict code execution results. This paper tries to simply the problem with a simplified programming language and generate code snippets in this language synthetically. The paper also investigates data size and model size scaling.

**Weaknesses:**

My main concern is the complexity of the code snippets generated by TinyPy Generator. Does the generator generates sufficiently complex logic, algorithms and data structures that mimics the complexity of today's code? If it is not close, it is not that representative and limits the value of the research. And the fact that 16M model achieves 99.13% accuracy indicates that the complexity of those code snippets is not high.
Another is that today LLMs are at the level of tens and hundreds billion of parameters. The capability these LLMs can achieve is not that comparable with old-fashion tens/hundreds of million parameters. This also makes the paper weaker in pushing forward the understanding of LLMs for code execution.

**Questions:**

- Why don't you conduct an experiment to finetune CodeLlama on the training dataset same as TEX? I hypothesize that you will get a comparable accuracy number as TEX.
- Could you explain the complexity of the generated code? Does the generated code match the sufficiently complex logic, algorithms and data structures that mimics the complexity of today's code?

---

> ### Author Response · Authors · 2024-11-24
>
> We sincerely appreciate your thoughtful review and insightful questions. Your feedback is valuable in helping us improve our work and clarify our contributions. We'd like to address your concerns and questions as follows:
> 1. Regarding fine-tuning CodeLlama:
> Fine-tuning CodeLlama on our dataset is outside the scope of our study, which focuses specifically on the capabilities of tiny language models trained from scratch. Our comparison with existing LLMs in Table 1 was primarily to demonstrate that our test set is non-trivial, as even large models struggle with it. The purpose of our work is not to compete with or improve upon large language models, but to explore the fundamental capabilities of small models in code execution tasks.
> This approach allows us to investigate an important question: can we achieve high performance on code execution tasks with significantly smaller models and less computational resources? Our results suggest that this is indeed possible, which could have important implications for efficient AI deployment in resource-constrained environments.
> 2. Complexity of the generated code:
> We appreciate your concern about the complexity of our generated code. Our generated code, while simplified, captures fundamental programming concepts including variable assignments, arithmetic operations, conditional statements, and loop constructs. While it doesn't match the full complexity of production-level software, it does represent core computational logic that forms the basis of more complex algorithms.
> The challenge posed by our test set is evidenced by the struggles of large language models in accurately predicting outputs, as shown in Table 1. Our focus was on creating a dataset that is complex enough to challenge models while remaining within the scope of what tiny models can potentially learn, allowing us to explore fundamental code execution capabilities.
> It's important to note that our goal was not to replicate the full complexity of modern software systems, but rather to isolate and study the core ability of language models to execute code. The high accuracy achieved by our small model on this task is significant precisely because it demonstrates that fundamental code execution capabilities can be learned by much smaller models than previously thought.
> We believe this finding opens up new avenues for research into the scalability and efficiency of language models in code-related tasks. It suggests that for certain fundamental coding tasks, we may not need the massive models that are currently the focus of much research, which could have significant implications for the accessibility and deployment of AI in coding applications.
> Thank you again for your valuable feedback. We hope these clarifications address your concerns and highlight the significance of our work in advancing the understanding of language model capabilities in code execution tasks.

---

### Official Review · Reviewer_eke6 · 2024-11-03

**Soundness:** 2
**Presentation:** 3
**Contribution:** 1
**Rating:** 3
**Confidence:** 4

**Summary:**

This paper explores whether small language models, trained from scratch with a simple next-token prediction objective, can learn to execute code effectively. Despite advancements in large language models (LLMs), their limitations in code execution remain unclear. To investigate, the authors create TinyPy, a dataset contains small programs with basic arithmetic and control flow operations. Using millions of randomly generated TinyPy code snippets and their outputs, they train various tiny language models on the next-token prediction task. Their proposed model, TEX, with only 15M parameters, achieves a high 99.13% accuracy in predicting code outputs, and outperforms various LLMs such as CodeLlama and GPT-4o.

**Strengths:**

1. This paper studies an interesting topic, the prediction of program's execution states.

**Weaknesses:**

1. Lack of novelty. This paper proposes a dataset TinyPy, yet the dataset is much simpler compared to practical programs (such as MBPP) as it only supports arithmetic operations, condition, and loops. Does it supports data structure such as List? The TEX model's architecture is also standard GPT-2, so essentially it is simply training a small GPT-2 on synthesized simple programs.
2. The results of comparison against other LLMs is not surprising, as the other LLMs (such as CodeLlama and GPT-4o) are conducted with one-shot prompting, while the TEX model is well-trained using 64M samples. Given that the TinyPy programs are simple and limited, such large amount of training data should be enough to well-capture the patterns and thus easy for TEX to learn. Although TEX shows high accuracy on the test set, it is unconvincing to show that the approach can be generalized to more complex programs. To better understand the comparison against larger LLMs, it could be useful to add additional experiments such as fine-tuning the larger models on the TinyPy dataset or evaluating TEX in a few-shot setting.

**Questions:**

1. Can this approach be generalized to more complex programs or real-world programs? To generalize to more complex programs, does it require synthesizing a training dataset for more complex programs? Is it feasible to synthesize such a complex/real-world programs dataset automatically?
2. Could the author clarify the novelty and contribution of the approach? This is unclear since synthesizing programs using defined grammars is not new, and fine-tuning a GPT-2 model is also not new. What are the implication of this approach to future works?

---

> ### Author Response · Authors · 2024-11-24
>
> We sincerely appreciate your thorough review and insightful questions. We'd like to address your concerns and clarify some key points about our work:
>
> Firstly, we want to emphasize that the goal of our study is not to challenge or outperform state-of-the-art large language models. Rather, our aim is to explore the fundamental capabilities of tiny language models in code execution tasks. The results presented in Table 1 serve primarily to demonstrate that our test set is non-trivial, as even large language models trained on billions of code tokens struggle to achieve perfect results on it.
>
> Regarding your specific questions:
>
> 1. Generalization to more complex programs:
> While our current approach focuses on a simplified Turing-complete language, the principles demonstrated could potentially be extended to more complex programs. Generalizing to real-world programs would indeed require synthesizing more sophisticated training datasets, which presents both challenges and opportunities for future research. While automatically generating datasets that fully capture real-world program complexity is currently a significant challenge, our approach of using a defined grammar could be incrementally expanded to incorporate more advanced language constructs, serving as a stepping stone towards handling more complex code execution tasks.
>
> 2. Novelty and contribution of the approach:
> Our work's novelty lies in demonstrating that a tiny language model of only 15M parameters, trained from scratch (not fine tuned) using next-token prediction, can achieve 99.13% accuracy in executing code from a Turing-complete language. This is significant as it shows that even small models can learn complex tasks like code execution without relying on pre-existing knowledge from larger pre-trained models. Our study provides concrete evidence that fundamental code execution capabilities can be achieved with much smaller models than previously thought, challenging assumptions about the necessary scale for such tasks. This finding has implications for the development of more efficient, resource-conscious AI systems for code-related tasks, particularly in environments where computational resources are limited.
>
> We believe these findings contribute valuable insights to the field, particularly in understanding the potential of small, efficient models for code-related tasks. We appreciate your feedback and hope this clarification addresses your concerns.
>
> Thank you for your time and consideration.

---

> > ### Comment · Reviewer_eke6 · 2024-12-02
> >
> > Thanks for the response.
> >
> > I understand that the synthesized code in your experiments captures the fundamental structures of a programming language, such as operations, loops, and conditions. However, real-world code is significantly more complex in terms of semantics. It often includes a diverse set of statements, deeply nested structures (e.g., loops within conditions, conditions within loops, and nested loops), and more sophisticated data structures (user-defined class, struct, etc.). While the conclusion that tiny LLMs can learn the execution of code with the level of complexity in your experiments is valid, it is not particularly surprising and is within my expectations.
> >
> > I agree with reviewer XpA7 that the contribution, as presented, does not meet the threshold for conferences like ICLR. A more impactful research direction could involve increasing the complexity of the code to include real-world scenarios, such as LeetCode solution programs, and evaluating LLMs' performance. It would be insightful to explore solutions for improving LLMs' understanding and execution capabilities on such complex, real-world programs, where achieving 99% accuracy would likely be much more challenging.

---

### Official Review · Reviewer_8aDH · 2024-11-04

**Soundness:** 2
**Presentation:** 3
**Contribution:** 2
**Rating:** 5
**Confidence:** 4

**Summary:**

The paper explores the potential of small language models (under 20 million parameters) in code execution tasks, typically a challenging area for larger language models. The authors trained a 15M-parameter model, TEX, using only the next-token prediction objective. With a dataset of randomly generated TinyPy code snippets, TEX achieves an impressive significant high accuracy in executing TinyPy code. This study suggests that tiny models can handle code execution tasks, provided they are given suitable training data and a simplified language structure.

**Strengths:**

1. **Innovative Use of Tiny Models**: This study makes a valuable contribution by demonstrating that, with proper training, small models can achieve near-perfect accuracy on structured tasks like code execution. This finding challenges the prevailing notion that only large models are capable of such tasks.

2. **Significant Efficiency Implications**: The success of small models like TEX in handling specific tasks could reduce the dependency on large, resource-intensive models in certain applications, paving the way for more efficient and scalable model deployment.

3. **Clear Presentation**: The paper is well-written and easy to follow, making the methodology and findings accessible to readers.

**Weaknesses:**

1. **Unclear Study Motivation**: Although the paper evaluates the code execution capabilities of a neural network model, it is not clear why this capability is necessary, given the existing functions of compilers and interpreters. One possible motivation might be that LLMs could serve as lightweight predictors of execution outcomes without actually running the code. However, the paper lacks any evaluation results to support this assumption.

2. **Limited Evaluation Setting**: The model is trained and evaluated on the same distribution, which may bias the results in favor of the proposed model over baseline models like GPT. For a more robust evaluation, it would be helpful if the authors tested the model on an uncontaminated dataset [1, 2]. Additionally, the benchmark used in this study seems overly simplified compared to real-world programming tasks, and testing the model on more realistic benchmarks could better demonstrate its effectiveness.

3. **Limited Novelty**: The study presents promising results, but the technical contributions are minimal. Both the model architecture and learning algorithm are widely studied in prior work. The promising outcomes observed might be attributed to data contamination or the evaluation setup rather than any novel technique introduced by this paper.


[1] GSM-Symbolic: Understanding the Limitations of Mathematical Reasoning in Large Language Models

[2] PPM: Automated Generation of Diverse Programming Problems for Benchmarking Code Generation Models

**Questions:**

1. Can the proposed approach still outperform the baseline if the baseline is fine-tuned on the dataset?

2. What is the motivation for using an LLM to predict code execution outcomes instead of leveraging existing compilers?

---

> ### Author Response · Authors · 2024-11-24
>
> We sincerely appreciate your thorough review and insightful comments. Before addressing your specific questions, we'd like to address some of the weaknesses you've pointed out:
>
> Regarding the study motivation and novelty, our work aims to challenge the prevailing notion that only large models can handle complex, structured tasks like code execution. Our study demonstrates that a tiny language model of only 15M parameters, trained from scratch using next-token prediction, can achieve 99.13% accuracy in executing code from a Turing-complete language. This finding is significant as it shows that fundamental code execution capabilities can be learned by much smaller models than previously thought, potentially reducing computational requirements for certain coding tasks.
>
> We acknowledge the limitations in our evaluation setting and appreciate your suggestions for improvement. Future work could indeed involve testing on uncontaminated datasets and more complex, real-world programming tasks to further validate our findings.
>
> Now, to address your specific questions:
>
> 1. Regarding fine-tuning baselines on our dataset:
> Our study's primary focus was on exploring the capabilities of tiny language models trained from scratch, rather than comparing with fine-tuned larger models. While fine-tuning larger models on our dataset could potentially improve their performance, it's outside the scope of our current research. Our goal was to demonstrate that small models can learn code execution effectively without relying on pre-existing knowledge from larger pre-trained models. This approach allows us to isolate and study the fundamental learning process of code execution in resource-constrained settings.
>
> 2. Motivation for using LLMs instead of existing compilers:
> Our motivation is not to replace existing compilers, but to investigate the fundamental capabilities of language models in understanding and processing code. This research provides insights into how AI models interpret and reason about programming constructs, which could inform future developments in code-related AI tasks such as code generation, completion, and debugging. Additionally, studying code execution in language models helps us understand their limits and potentials in handling structured, logical tasks, which has broader implications for AI's ability to perform complex reasoning.
>
> We believe these insights contribute valuable knowledge to the field, particularly in understanding the potential of small, efficient models for code-related tasks. We appreciate your feedback and hope this clarification addresses your concerns.
>
> Thank you for your time and consideration.

---

> > ### Comment · Reviewer_8aDH · 2024-12-02
> > **Review comments**
> >
> > Thank you for your explanation.
> >
> > I understand that the primary focus of this work was to explore the capabilities of tiny language models trained from scratch. However, I believe it would also be valuable to conduct an experiment fine-tuning a larger pre-trained language model (LLM) to highlight the advantages of training a small model from scratch.
> >
> > Regarding the motivation for this work, I agree that it has the potential to provide meaningful insights into how AI models interpret and reason about programming constructs. The key question, however, is what specific insights will be derived and how these findings can be leveraged to further improve code models. I suggest the authors put more effort into clarifying the relationship between "predicting code execution" and enhancing the performance of code models.
> >
> > Consider both the pros and the cons of this paper, I will keep my original score.

---

### Meta-Review · Area_Chair_ScWG · 2024-12-08

**Metareview:**

The paper investigates the capabilities of small language models (under 20 million parameters) in executing code, a task typically challenging for larger models. The authors trained a 15M-parameter model named TEX using a next-token prediction objective on a dataset of randomly generated TinyPy code snippets. TEX demonstrated impressive accuracy in executing these snippets, suggesting that small models can effectively handle code execution tasks with appropriate training data and simplified language structures.

All the reviewers raise concerns over the motivation, evaluation rigor, and novelty of the paper. Addressing these issues through more experiemnts would be essential for a stronger contribution to the field.

**Additional Comments On Reviewer Discussion:**

All reviewers agree that while the paper presents interesting findings, it suffers from significant weaknesses in motivation, evaluation rigor, and novelty.

---

### Decision · Program_Chairs · 2025-01-22

Reject